# HEALNet: Multimodal Fusion for Heterogeneous Biomedical Data

**Konstantin Hemker**
Department of Computer Science & Technology
University of Cambridge
Cambridge, United Kingdom
konstantin.hemker@cl.cam.ac.uk

**Nikola Simidjievski**
PBCI, Department of Oncology
University of Cambridge
Cambridge, United Kingdom
ns779@cam.ac.uk

**Mateja Jamnik**
Department of Computer Science & Technology
University of Cambridge
Cambridge, United Kingdom
mateja.jamnik@cl.cam.ac.uk

## Abstract

Technological advances in medical data collection, such as high-throughput genomic sequencing and digital high-resolution histopathology, have contributed to the rising requirement for multimodal biomedical modelling, specifically for image, tabular and graph data. Most multimodal deep learning approaches use modality-specific architectures that are often trained separately and cannot capture the crucial cross-modal information that motivates the integration of different data sources. This paper presents the **H**ybrid **E**arly-fusion **A**ttention **L**earning **Net**work (HEALNet) – a flexible multimodal fusion architecture, which: a) preserves modality-specific structural information, b) captures the cross-modal interactions and structural information in a shared latent space, c) can effectively handle missing modalities during training and inference, and d) enables intuitive model inspection by learning on the raw data input instead of opaque embeddings. We conduct multimodal survival analysis on Whole Slide Images and Multi-omic data on four cancer datasets from The Cancer Genome Atlas (TCGA). HEALNet achieves state-of-the-art performance compared to other end-to-end trained fusion models, substantially improving over unimodal and multimodal baselines whilst being robust in scenarios with missing modalities. The code is available at https://github.com/konst-int-i/healnet.

## 1 Introduction

A key challenge in Multimodal Machine Learning is *multimodal fusion*, which is the integration of structurally heterogeneous data into a common representation that reduces the dimensionality of the data whilst preserving salient biological signals [Steyaert et al., 2023]. Fusion approaches are well-studied in areas where there is a clearly defined shared semantic space, such as audio, visual, and text tasks like visual question answering [Goyal et al., 2016], image captioning [Yu et al., 2020], or multimodal dialogue [Liang et al., 2023]. However, healthcare data commonly consists of 2D or 3D images (histopathology and radiology), graphs (molecular data), and tabular data (multi-omics, electronic health records), where cross-modal relationships are typically more opaque and complex, the modalities often do not share semantics, and common representations are less explored. The

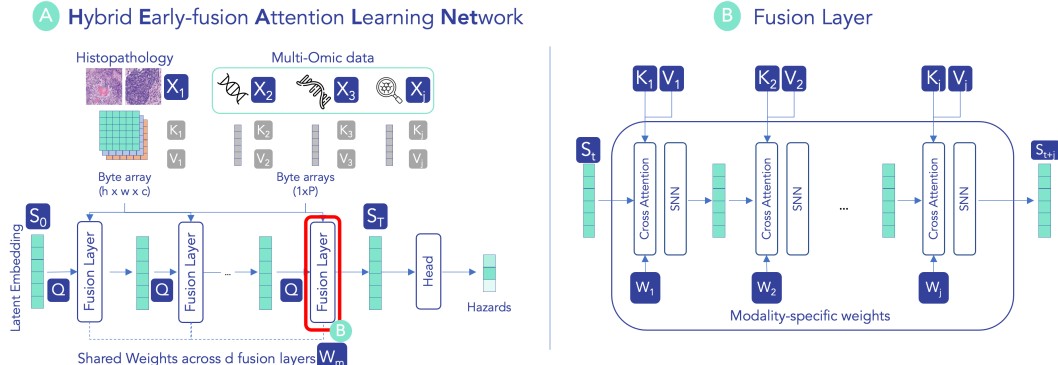

Figure 1: Overview of HEALNet (**H**ybrid **E**arly-fusion **A**ttention **L**earning **Net**work) using a shared *and* modality-specific parameter space to learn from structurally different data sources in the same model (Fig. 1A). The shared space is a learned latent embedding $S$ that is iteratively updated through $d$ attention-based fusion layers and captures the shared information between modalities. The hybrid early-fusion layer (Fig. 1B, and Eq. 3) learns the cross-attention weights $W_m = \{W_m^{(q)}, W_m^{(k)}, W_m^{(v)}\}$ for each modality $m$ corresponding to the queries ($Q_m = W_m^{(q)} S$), keys ($K_m = W_m^{(k)} X_m$), and values ($V_m = W_m^{(v)} X_m$) which are shared between layers. These layers capture the structural information of each modality and encode it in the shared embedding after a pass through a self-normalising network (SNN) layer.

*fusion stage* describes how far the multimodal representation is removed from the raw (unimodal) data, and is commonly categorised into early, intermediate, and late fusion [Baltrusaitis et al., 2019].

Early fusion approaches combine the raw data early in the pipeline, which allows the training of a single model from all data modalities simultaneously. However, most of these approaches use simple operations such as concatenation, which removes structural information, or take the Kronecker product [Chen et al., 2022a], which can lead to exploding dimensions when applied to multiple modalities and large matrices. Late fusion, on the other hand, trains separate models for each modality at hand, which allows capturing the salient structural information but prevents the model from learning interactions between modalities [Liang et al., 2023]. Intermediate fusion approaches attempt to overcome this trade-off by learning a low-level representation (embedding) for each modality before combining them. This can result in discovering the cross-modal interactions whilst taking advantage of each modality's internal data structure. The problem with many intermediate fusion approaches is that the learnt latent representation is not interpretable to human experts, and handling missing modalities is often noisy [Cui et al., 2023]. To overcome these issues, we posit that there is a need for more sophisticated early fusion methods, which we refer to as *hybrid early-fusion*, that: a) preserve structural information *and* b) learn cross-modal interactions, and c) work on the raw data, thus allowing for in-model explainability.

In this paper, we propose **H**ybrid **E**arly-fusion **A**ttention **L**earning **Net**work (HEALNet, Figure 1), a novel *hybrid early-fusion* approach that leverages the benefits of early and intermediate fusion approaches, and scales to any number of modalities. The main idea behind HEALNet is to use both a shared and modality-specific parameter space in parallel within an iterative attention architecture. Specifically, a shared latent bottleneck array is passed through the network and iteratively updated, thus capturing shared information and learning tacit interactions between the data modalities. Meanwhile, attention weights are learned for each modality and are shared between layers to learn modality-specific structural information. We demonstrate the multimodal utility of HEALNet on survival analysis tasks on four cancer sites from The Cancer Genome Atlas (TCGA) data, combining multi-omic (tabular) and histopathology slides (imaging) data. Our results show that HEALNet achieves state-of-the-art concordance Index (c-Index) performance compared to other fusion models on all four cancer datasets for multimodal patient survival prediction. More specifically, HEALNet leads to an average improvement of up to 7% compared to the best unimodal benchmarks and up to 4.5% compared to the best early, intermediate, and late fusion benchmarks, which we see as a promising validation of our hybrid early-fusion paradigm. In summary, the contributions of our proposed HEALNet include:

- *Preserving the modality-specific structure*: HEALNet outperforms unimodal tabular (omic) and imaging (histopathology) baselines without a dedicated modality-specific network topology.

- *Learning cross-modal interactions*: HEALNet effectively captures cross-modal information, achieving a significantly higher multimodal uplift compared to existing early, intermediate, and late fusion baselines.

- *Handling missing modalities*: We show that HEALNet effectively handles missing modalities at inference time without introducing further noise to the model, a common problem in the clinical use of multimodal models.

- *Model inspection*: HEALNet is explainable 'by design' since the modality-specific attention weights can provide insights about what the model has learned without the need for a separate explanation method. We believe that they are useful for model debugging and validation alongside domain experts.

## 2 Related Work

In this paper, we focus on multimodal learning problems from biomedical data, where the data modalities are *structurally heterogeneous*, specifically combining image (e.g., Whole Slide Imagery (WSI)) and tabular (e.g., omic and clinical) data. This aspect is different and more general than approaches that focus on combining homogeneous modalities, such as multi-omic data, where the combined modalities have the same structural formalism (tabular) [Sammut et al., 2021, Dai et al., 2021]. As such, our work closely relates to several approaches for multimodal data fusion that consider learning from WSI images and genomic data. Namely, Cheerla and Gevaert [2019] introduce a two-step procedure, combining a self-supervised pre-training step with a downstream fine-tuning for survival analysis. In the first self-supervised step, a modality-specific embedding is trained and optimised using a similarity loss (similar to contrastive learning) between the embeddings. The latter step includes (supervised) fine-tuning of a survival model, trained and optimised using the Cox loss. HEALNet, on the other hand, implements a sequential architecture for end-to-end training to learn across modalities without requiring a separate pre-training step. The model architecture is flexible to allow for both training on the raw input data as well as leveraging pre-trained model encoders, which may further improve its performance on a particular task of interest.

HEALNet's multimodal fusion capabilities build on attention architectures [Vaswani et al., 2017]. A popular such architecture is the Perceiver [Jaegle et al., 2021], which uses iterative self- and cross-attention layers and achieves impressive performance across various unimodal tasks. However, this architecture is restricted to single modalities unless inputs are concatenated before training, which may remove salient structural signals. In a multimodal context, cross-attention has been used as a core component for several intermediate and late fusion models. 'Multimodal co-attention' (MCAT) [Chen et al., 2021] is an attention-based fusion approach that uses a tabular modality as the query and the imaging modality as the key and value array to train a cross-attention unit. Xu and Chen [2023] further extend this concept by introducing a refined co-attention mechanism based on optimal transport, where one modality is used to better contextualise the other. Such co-attention approaches only scale to two modalities since the co-attention units can only take in one set of query-key-value inputs. It also requires having a 'primary' modality that should be contextualised by a 'secondary' modality, which may not always be the case.

Chen et al. [2022a] present an intermediate fusion approach that first constructs modality-specific embeddings before passing them through a gating-based attention mechanism, combining the output via a Kronecker product. In turn, the resulting high-dimensional '3D multimodal tensor' is used for a downstream survival prediction task. Similarly, Chen et al. [2022b] propose a late fusion approach which implements modality-specific model encoders before combining each model's output via a gating mechanism. Here, the attention mechanism is only applied to the imaging modality (WSI), which serves as a method for learning more general representations by combing the patch-level latents, but also allows for explainable post-hoc image analysis of the identified regions. An inherent limitation of such an approach is that the encoders are trained separately (without the other modality's context), and consequently, the explanations only account for unimodal information.

Another common limitation of all the above approaches is that they expect fixed tensor dimensions during training and inference. An alternative to this is presented with sequential fusion methods such as by Swamy et al. [2023], where modalities can be skipped if not present. Currently, the problem

with this approach is that it only works on 1D tensors and latent states, relying on encoders specific to this architecture.

In contrast, HEALNet overcomes many of the limitations of the related fusion approaches. First, its design readily scales to more than two modalities without additional computational overhead beyond the one introduced by the additional modality-specific encoders. Namely, the iterative attention mechanism alleviates the use of fusion operators (such as a Kronecker product) that render high-dimensional embeddings, and allows for combining many modalities while preserving the structural information of each. Moreover, HEALNet can learn cross-modal interactions since the modalities are used as mutual context for each other's updates. This also allows for more holistic explanations compared to late fusion approaches that use modality-specific models trained in isolation.

## 3 HEALNet

**Preliminaries.** Let $X_m$ represent data from modality $m = 1, ..., j \in \mathbb{N}$. Let $X_m \in \mathbb{R}^{p \times n}$ be either a tabular dataset with $p$ features and $n$ samples or an image dataset $X_m \in \mathbb{R}^{h \times w \times c \times n}$ with $n$ images with height $h$, width $w$ and channels $c$. The goal of a multimodal fusion approach is to learn a fusion function $f()$ such that $y = f(X_1, ..., X_j)$. A conventional design of such a system (such as the related work discussed in Section 2) is to first learn a modality-specific function $g_m()$, which learns an intermediate representation $h_m = g_m(X_m)$, and then apply a fusion function $f()$ for predicting the target variable $\hat{y} = f(h_1, ..., h_j)$.

**Architecture.** We depict HEALNet in Figure 1. Instead of computing $h_m$ and applying a single fusion function $f()$, HEALNet uses an iterative learning setup. Let $t$ denote a step, where the total number of steps $T = d \times j$ for the number of fusion layers $d \in \theta$. Let $S_t$ represent a latent array shared across modalities, initialised at $S_0$ where $S \in \mathbb{R}^{a \times b}$ for embedding dimensions $a, b \in \mathbb{N}$ and is updated at each step. First, instead of learning an intermediate representation $h_m$ as encoded inputs for $X_m$, we compute the attention weights as

$$a_m^{(t)} = \alpha(S_t, X_m), \tag{1}$$

for each modality $m$ at each step $t$. Second, we learn an update function $\psi()$ to be applied at each step. The update of $S$ with modality $m$ is given by

$$S_{t+1,m} = \psi(S_t, a_m^{(t)}), \tag{2}$$

for total time steps $T$ and attention function $\alpha$ (Equation 1). For parameter efficiency, the final implementation uses weight sharing between layers. Across modalities, each early-fusion layer becomes an update function of the form

$$S_{t+j} = \psi(S_t, a_1, ..., a_j). \tag{3}$$

The final function for generating a prediction only takes the final state of the shared array and returns the predictions of the target variable $\hat{y} = f(S_T)$ as a fully-connected layer.

Figure 1 depicts a high-level visual representation of this approach, showing: (a) Hybrid Early-fusion Attention Learning Network, and its key component (b) the early fusion layer (as given in Equation 3). We use attention layers since they: a) make fewer assumptions about the input data (e.g., compared to a convolutional network), and b) their ability to provide context to the original modality through the cross-attention mechanism. We start by initialising a latent embedding variable, which is iteratively used as a query into each of the fusion layers, and is updated with information from the different modalities at each layer pass. We chose the iterative attention paradigm due to its highly competitive performance on a range of unimodal tasks [Jaegle et al., 2021]. Passing the modalities through the shared latent array helps to significantly reduce the dimensionality whilst learning important structural information through the cross-attention layers. The HEALNet pseudocode is detailed further in Appendix A.

**Preserving structural information.** To handle heterogeneous modalities, we use modality-specific cross-attention layers $\alpha()$ and their associated attention weights $a_m^{(t)}$, whilst having the latent array $S$ shared between all modalities. Sharing the latent array between modalities allows the model to learn from information across modalities, which is repeatedly passed through the model (Figure 1A). Meanwhile, the modality-specific weights between the cross-attention layers (Figure 1B) focus on

learning from inputs of different dimensions, as well as learning the implicit structural assumptions of each modality. Specifically, in this work, the employed attention mechanism refers to the original scaled dot product attention from [Vaswani et al., 2017], with adjustments for tabular and image data.

Formally, given a tabular dataset as a matrix $X_m = \{x_m^{(11)}, ..., x_m^{(np)}\}$, with $n \in N$ samples and $p \in P$ features (e.g., a gene expression), we aim to learn the weight matrices $W_m^{(q)}$, $W_m^{(k)}$, and $W_m^{(v)}$ that act as a linear transformation for $S$ and $X_m$ that form the queries ($q_m^{(n)}$), keys ($k_m^{(n)}$) and values ($v_m^{(n)}$) for each sample passed into the layer. The general scale dot-product attention generates attention scores for each feature and can be expressed in Cartesian Notation as

$$\alpha(q_p, K) = \sum_{i=1}^{P} \left[ \frac{exp(q_p \cdot k_i^P)}{\sum_j exp(q_p \cdot k_j^P)} \right] \quad \forall j \in [1, N]. \tag{4}$$

In other words, for each channel $p$ and sample $n$, an attention layer calculates the normalised and scaled attention weight being given the context of all other features for that sample. This has the benefit that the attention scores are always specific to each input given to the attention layer. From this, we can extract both the normalised attention matrix $A$ as well as the context matrix $C_p(q, K, V) = \sum_{i=1}^{P} A_{p,i} \times v_i$, which is the attention-weighted version of the original input $x$. In our case, we need to combine multiple inputs to apply the iterative attention mechanism (i.e., cross-attention) – these inputs are the latent $S$ and the input matrix $X_m$ for each modality. To do this, we use the latent array as the query and the input tensor as the keys and values, respectively. Given a latent array $S$, we define the query for each sample $q_m^{(n)} = W_m^{(q)} S$ and the keys and values as $k_m = W_m^{(k)} x$ and $v_m^{(n)} = W_m^{(v)}$ for all samples $n \in [1, N]$. Intuitively, the iterative cross-attention can be seen as aligning the query to each modality individually, but not aligning the modalities themselves to ensure that its unique signal is captured. At each time step, the query for the next update provides context from the other modalities of previous updates.

**High-dimensional biomedical data.** Attention-based architectures are typically trained on vast datasets (which are commonly available for vision and language tasks). The challenges of working with biomedical data, however, are their high dimensionality whilst often having relatively few samples (i.e., patients). For example, a dataset (such as TCGA-BLCA) contains whole slide images of approximately 6.4 gigapixels (80k × 80k pixels) in its highest resolution and includes thousands of multi-omic features, but only from a few hundred patients in total. This leads to two common problems in digital pathology – overfitting [Holste et al., 2023] and high computational complexity.

First, to counteract overfitting, HEALNet implements both L1 and L2 regularisation. Considering the relatively large number of parameters required for the attention layers, we found L1 regularisation to be important. Beyond that, we opted for a self-normalising neural network (SNN) block, due to its proven robustness and regularisation properties [Klambauer et al., 2017].

Second, handling the extremely high resolution of the whole slide images (WSIs) within computational constraints is also a challenge. We address this by extracting non-overlapping 256x256 pixel patches on the 2x and 4x downsampled whole-slide image (~0.5 and $1.0\mu m$ per pixel respectively). For comparability with other work, we extract a 2048-dimensional feature vector for each patch using a standard ResNet50 pre-trained on the Kather100K dataset, which consists of 100k histopathology images of both healthy tissue and colorectal cancer tissue [Pocock et al., 2022]. While HEALNet also achieves competitive results on the raw patch data, this requires more significant downsampling to be computationally feasible at scale.

**Handling missing modalities.** A common challenge in clinical practice is missing data modalities during inference. Namely, in practical scenarios, while models have been trained on multiple modalities, there is a great chance that not all data modalities are available for predicting the patient's outcome. Therefore, multimodal approaches must be robust to such scenarios. Typical intermediate fusion approaches would need to randomly initialise or impute a tensor of the same shape, or sample the latent space for a semantically similar replacement to pass into the fusion function $f(h^1, ..., h^j; \theta)$ at inference, which is likely to introduce noise. In contrast, HEALNet overcomes this issue by design: the iterative paradigm can simply skip a modality update step (Equation 3) at inference time in a noise-free manner. Note that these practical benefits also extend to training scenarios, where a (typically small) number of samples are missing some modalities. Rather than imputing this data or completely omitting the samples, HEALNet can train and utilise all the available data using the same update principle.

# 4 Experiments

**Datasets.** We empirically evaluate the utility of HEALNet on survival analysis tasks on four cancer datasets from The Cancer Genome Atlas (TCGA). Concretely, we use modalities which are structurally heterogeneous such as the ones formalised in a tabular or image dataset. Our tabular data structures consist of three sources: bulk gene expressions (RNAseq), mutations (whole-genome sequencing), and copy number variations. HEALNet treats these as three separate modalities, while for the baselines that only support two modalities we had to concatenate them – in continuation of this paper we refer to these as *omic modality*. The *WSI* modality includes H&E-stained whole slide tissue images of the same patient cohorts as in the omic modality. Namely, the four cancer datasets that we include are Muscle-Invasive Bladder Cancer (BLCA, n=436), Breast Invasive Carcinoma (BRCA, n=1021), Cervical Kidney Renal Papillary Cell Carcinoma (KIRP, n=284), and Uterine Corpus Endometrial Carcinoma (UCEC, n=538) (further dataset details are provided in Appendix C). These specific sites were chosen based on their sample size (BRCA, BLCA, and UCEC are some of the largest TCGA datasets), performance indicators reported in previous unimodal studies (e.g., KIRP highest on omic, UCEC highest on WSI only [Chen et al., 2022b]), and other omic properties (e.g., BLCA and UCEC are known for their very high gene mutation rate [Ma et al., 2022]).

**Task setup.** We focus on modelling overall survival. For each patient, we are provided with right-censored censorship status $c$ (a binary variable on whether the outcome was observed at the end of the study or not) and survival months since data recording. In line with our baselines [Chen et al., 2022b], we take non-overlapping quartiles $k$ for the uncensored patients and apply them to the overall population, which assigns a categorical survival risk to each patient. For the survival task, our prediction model and the baselines are set to output logits of these survival buckets $y_{logits} = f(X_m, S; \theta, \rho)$. Using these, we calculate the hazard as the sigmoid of the logits $f_{hazard} = \frac{1}{1e^{-y_{logits}}}$, and the corresponding survival as $f_{survival} = \prod_1^k 1 - f_{hazard}$. The survival, hazards, censorship status and discretised survival label are all used to calculate the negative log-likelihood (NLL) loss from the proportional hazards model defined in [Zadeh and Schmid, 2021]. The concordance index is then calculated by comparing all study subject pairs to determine the fraction of pairs in which the predictions and outcomes are concordant [Brentnall and Cuzick, 2018].

During development, we found that the proportion of uncensored patients sometimes can be as little as 15% of the cohort (UCEC). Applying the survival bins from such a small sub-sample led to very imbalanced discretised survival on the full cohort. To counteract this, we added the option to apply survival class weighting in the loss function, implemented as the inverse weight of the survival bins. Additionally, note that the NLL loss and the concordance index are sometimes only loosely related. Therefore, our loss weighting helped to stabilise the correlation between the NLL loss and the c-Index. To ensure fair comparison and comparability with the baselines, we employed the same NLL loss and weighting for training HEALNet. Nevertheless, HEALNet is readily extensible and can be implemented with other survival loss functions such as the Cox loss, which can potentially lead to more stable c-Index results [Cheerla and Gevaert, 2019].

**Baselines.** In all experiments, we compare HEALNet to state-of-the-art uni- and multimodal approaches that utilise different fusion strategies. This includes Porpoise [Chen et al., 2022b], which uses a *late fusion* gating method to combine the latent representations learned from a self-normalising network (SNN) on the omic modality and an attention-based multiple-instance learning network (AMIL) [Chen et al., 2022b]. In terms of *intermediate fusion*, we include MCAT [Chen et al., 2021] and MOTCAT [Xu and Chen, 2023], which leverage co-attention between latent representations resulting from modal-specific encoders. Furthermore, we compare to MultiModN [Swamy et al., 2023], a *sequential fusion approach* for multi-task learning with a pre-defined set of encoders. We also include the Perceiver model [Jaegle et al., 2021], which has shown strong performance on various unimodal tasks through its iterative attention mechanism. Following the original implementation, we benchmark its multimodal integration capabilities using *early fusion* via concatenation. Finally, for each of our multimodal baselines, we trained unimodal variants as reported in the task-specific papers [Chen et al., 2022b, 2021], as well as unimodal models trained with HEALNet using a single modality. We report the best unimodal model out of the set. The full results of all unimodal baselines can be found in Appendix B.

**Implementation details.** For each experiment we employ 5-folds of repeated random sub-sampling (Monte Carlo cross-validation) with a 70-15-15 split for the training, validation and test sets. All reported results show the models' performance on test data that was not used during training or

Table 1: Mean and standard deviation of the concordance Index on four survival risk categories. We trained HEALNet and all baselines on four TCGA tasks and report the performance on the test set across five folds. HEALNet outperforms all of its multimodal baselines and three out of four unimodal baselines in absolute c-Index performance.

| Model | BLCA | BRCA | KIRP | UCEC |
|---|---|---|---|---|
| Uni-modal (Omics) | $0.606 \pm 0.019$ | $0.580 \pm 0.027$ | $0.780 \pm 0.035$ | $0.550 \pm 0.026$ |
| Uni-modal (WSI) | $0.556 \pm 0.039$ | $0.550 \pm 0.037$ | $0.533 \pm 0.099$ | $\mathbf{0.630} \pm 0.028$ |
| Porpoise (Late) | $0.620 \pm 0.048$ | $0.630 \pm 0.040$ | $0.790 \pm 0.041$ | $0.590 \pm 0.034$ |
| MCAT (Intermediate) | $0.620 \pm 0.040$ | $0.589 \pm 0.073$ | $0.789 \pm 0.087$ | $0.589 \pm 0.062$ |
| MOTCAT (Intermediate) | $0.631 \pm 0.051$ | $0.607 \pm 0.069$ | $0.810 \pm 0.062$ | $0.587 \pm 0.083$ |
| MultiModN (Sequential Fusion) | $0.551 \pm 0.060$ | $0.582 \pm 0.084$ | $0.753 \pm 0.152$ | $0.610 \pm 0.121$ |
| Perceiver (Early Fusion) | $0.565 \pm 0.042$ | $0.566 \pm 0.068$ | $0.783 \pm 0.135$ | $0.623 \pm 0.107$ |
| HEALNet (ours) | $\mathbf{0.668} \pm \mathbf{0.036}$ | $\mathbf{0.638} \pm \mathbf{0.073}$ | $\mathbf{0.812} \pm \mathbf{0.055}$ | $\mathit{0.626} \pm \mathit{0.037}$ |

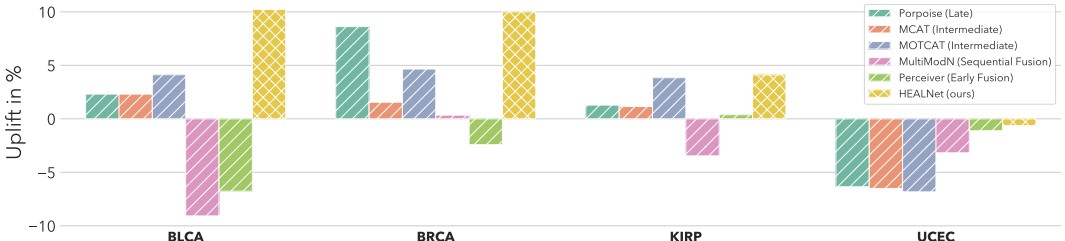

Figure 2: Mean percentage uplift of all multimodal models compared to the best unimodal baseline. Across all tested TCGA cancer sites, HEALNet's *hybrid early-fusion* paradigm outperforms early, intermediate, and late fusion methods.

validation. We re-train all of the baseline models using the code reported in the respective papers. All models have been run under the same circumstances and using the same evaluation framework (including data splits and loss weighting). For hyperparameter tuning, we ran a Bayesian Hyperparameter search [Bergstra et al., 2013] for all training parameters across models. Model-specific parameters of the baselines were tuned if the optimal parameters on the TCGA datasets were not available. The final set of hyperparameters can be found in Appendix D. All experiments were run on a single Nvidia A100 80GB GPU running on a Ubuntu 22.04 virtual machine. HEALNet is implemented in the PyTorch framework and available open-source at `https://github.com/konst-int-i/healnet`.

## 5 Results

The results of the survival analysis are summarised in Table 1, showing the mean and standard deviation of the c-Index across five cross-validation folds. Across all tested cancer sites, HEALNet outperforms all multimodal baselines. This corresponds to an improvement over the multimodal baselines of approximately 7%, 1%, 3% and 6% on the BLCA, BRCA, KIRP, and UCEC tasks respectively. HEALNet also exhibits more stable behaviour compared to the multimodal baselines, as evident from the standard deviation (across folds), which is lower in three out of four datasets.

The unimodal baselines shown in Table 1 correspond to the best-performing models from a selection of unimodal baselines we trained (refer to the complete list in Appendix B). Compared to the better of the two unimodal baselines, HEALNet achieves an approximately 10% higher c-Index on BLCA and BRCA, a 4% higher c-Index on KIRP, and a nearly equivalent performance on UCEC. We refer to this as a *multimodal uplift*, which is illustrated in Figure 2, where we compare the improvement of the different multimodal models and fusion strategies to the best unimodal model. Note that the UCEC dataset is an example of *modality dominance*, where all informative signal stems from one modality (in this case WSI), while the signal from the other modality can be either non-informative or noisy. Intermediate and late fusion approaches, which directly combine modalities, are less robust in such cases. For instance, in the case of Porpoise and MCAT, this can even lead to performance degradation. Since HEALNet is more robust to such noise, it leads to performance comparable to the unimodal variant.

Table 2: Analysis of the performance of HEALNet, trained on all modalities, in scenarios with missing modalities at inference, compared to unimodal baselines. Each test sample contains only one of either the Omic or WSI modality. The scenarios include test sets consisting of samples with only Omic modality, only WSI modality or a combination of both (at random). HEALNet achieves a higher c-Index across datasets, implying effective encoding of cross-modal information and handling different amounts of data with missing modalities.

| Test | 100% Omics | | 100% WSI | | 50%WSI + 50% Omics | | WSI+Omic |
|------|-----------|---------|-----------|---------|---------------------|---------|----------|
|      | Uni-modal | HEALNet | Uni-modal | HEALNet | Uni-modal | HEALNet | HEALNet |
| BLCA | 0.606 | 0.618 | 0.487 | 0.501 | 0.547 | 0.612 | **0.668** |
| BRCA | 0.556 | 0.571 | 0.529 | 0.539 | 0.543 | 0.541 | **0.638** |
| KIRP | 0.771 | 0.773 | 0.518 | 0.526 | 0.644 | 0.714 | **0.812** |
| UCEC | 0.509 | 0.529 | 0.558 | 0.584 | 0.533 | 0.580 | **0.626** |

To further assess the robustness of HEALNet, we evaluate its performance in scenarios with missing modalities. Specifically, using HEALNet trained on four modalities (WSI + 3×Omics), we investigate its performance when modalities are missing during inference. Note that half of the test samples include only a WSI modality, while the other half is an Omics modality, chosen randomly. The unimodal baseline corresponds to the predictions of the available modality in the same way that a late fusion model would use two unimodal models followed by an XOR gating mechanism to make its prediction. For completeness, we also report results where the whole test set consists of samples with either Omics or WSI modality, rather than a combination of both. Note that the unimodal baselines are HealNet models, trained on a single modality. The results from this analysis, given in Table 2, show that our proposed HEALNet, pre-trained on both modalities, archives stable and generally better performance than a late fusion baseline (as commonly performed in practice).

## 6   Discussion

**Structure-preserving fusion.** The results support our hypothesis that HEALNet is able to learn the structural properties of each modality and convert the structural signal into better performance. The quantitative evidence of this is given by HEALNet's absolute c-Index performance across tasks (Table 1), performing substantially better than the early fusion baseline that employs concatenation. Additionally, HEALNet allows for further qualitative analysis of this behaviour, as we can visualise the sample-level attention of different regions of the whole slide image. The attention maps in Figure 3 show a sample where the model identified multiple patches in the same region – distilling the wider image down to local information for which we would typically use convolutional networks.

HEALNet learns about high-level structural relationships by using a hybrid of modality-specific and shared parameters. For each modality, we learn attention weights (Equation 1) simultaneously in an end-to-end process. We believe that this distinguishes our hybrid early-fusion approach from conventional early and intermediate fusion methods. Instead of removing structural information (i.e., concatenation) or creating excessively large input tensors (i.e., Kronecker product), our hybrid early-fusion is able to preserve such structures by design. Similar to intermediate fusion methods, we use a shared latent space to capture cross-modal dependencies. However, instead of creating a latent space through multiple encodings before combining them via a downstream function ($f(h_1, ..., h_j; \phi)$), HEALNet learns an update function (Equation 3) that iteratively updates the shared latent with modality-specific information. Nevertheless, a limitation of such an approach is akin to training a higher number of parameters (i.e., large attention matrices), but on relatively few samples, making it prone to overfitting. Hence, we found that HEALNet can be sensitive to the choice of regularisation mechanisms, even though the current regularisation techniques (such as L1 + SNN) have shown to be effective (see Figure 4 in Appendix E).

**Cross-modal learning.** The motivation of using a hybrid early-fusion over a late fusion approach is to enable the model to learn cross-modal interactions that are unavailable to modality-specific models trained in isolation. We can see the effect of this in Figure 2, showing HEALNet's substantially higher uplift compared to the late fusion benchmark. Additionally, Table 2 shows that even when a modality is missing, HEALNet outperforms the best unimodal models that are present, indicating that the multimodal embedding is implicitly inferring *some* information about the missing modality.

We note, however, that using a multimodal model is not always a requirement, especially in the presence of *modality dominance*, which we see on the UCEC dataset. This dominance can be explained by the relatively high morphological variety that has been found in endometrial carcinomas (i.e., UCEC), which is expressed by spindled, stromal, or extremely large cells [Rabban, 2020]. This visual signal can only be picked up by the WSI, which is a trend we saw consistently across baselines. Nonetheless, HEALNet is robust to such cases, achieving comparable performance to the best unimodal model. Upon further inspection of the HEALNet's Omics attention weights on the UCEC task, we found that they barely changed since their initialisation. HEALNet was able to (correctly) inhibit this signal, which is not the case for the other multimodal baselines where it leads to a loss in performance.

**Missing modality handling.** A key benefit of using iterative attention is that we can skip updates if modalities are missing at inference time without adding additional noise. For many intermediate fusion methods, missing modalities introduce noise since the fusion function $f()$ expects an intermediate representation $h_m$ for all modalities. This requires initialising a random array or doing a latent search for a similar array to impute the missing portion. A practical approach to this challenge is a late fusion approach, which requires training and keeping several unimodal alternatives, that can act as a substitution. This, however, can be computationally intensive. HEALNet, on the other hand, overcomes this challenge by design. We believe that this underlines a key benefit of *hybrid early-fusion* – handling mixed missing modalities, at inference time, which takes advantage of multimodal training, without introducing additional noise.

**Computational complexity.** Another advantage of HEALNet's sequential architecture design is that it scales linearly with respect to both sample size $n$ and the number of modalities $m$. More specifically, since the cross-attention and self-normalising layers that comprise the fusion layers scale with $\mathcal{O}(n)$, each fusion layer has a complexity of $\mathcal{O}(mn)$. As a result, the runtime only depends on the number of fusion layers $d$, which in our setup is a hyperparameter (refer to Appendix D for the hyperparameter values used in this work). Compared to other competitive multimodal baselines, HEALNet scales much more efficiently. For instance, MCAT [Chen et al., 2021] is natively designed for two modalities. Scaling MCAT to $m > 2$ would require calculating the modality-guided cross-attention for all unique pairwise combinations, resulting in $\mathcal{O}(m^2n)$. This quadratic scaling trend is also similar for other baselines that implement Kronecker product for combining modalities, such as Porpoise [Chen et al., 2022b] and MOTCAT [Xu and Chen, 2023]. Note that, in all cases, including HEALNet, the actual runtime will further depend on several additional factors and designs, such as the choice of the modality-specific encoders and the size of their embeddings within the underlying multimodal model.

**Inspections and explanations.** Finally, another design benefit of using attention on the raw input data is that it allows for instance-level insights into the model's behaviour, without the need for additional post-hoc explanation methods. Figure 3 shows what parts of the sample the model attends to on average across layers. For images, one can create a high-level heatmap of the cell tissue to highlight relevant regions for more detailed insights into the tumour microenvironment and disease progression. In turn, these regions can be further analysed post-hoc, such as via nucleus segmentation. To showcase this capability, in Figure 3, we take the highest attention patches and perform nucleus segmentation

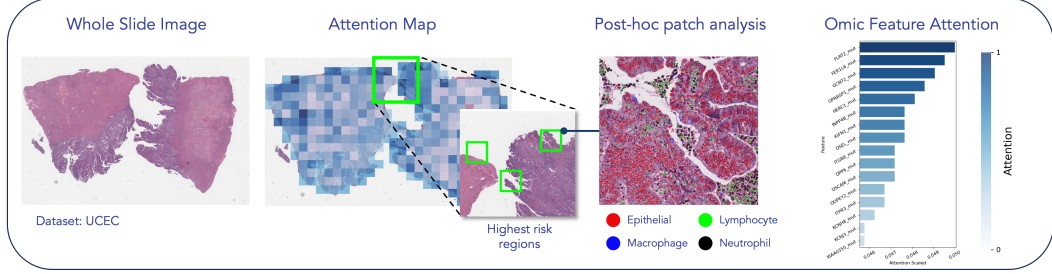

Figure 3: Illustration of model's inspection capabilities using HEALNet on a high-risk patient of the UCEC study. We use the mean modality-specific attention weights across layers to highlight high-risk regions and inspect high-attention omic features. Individual patches can be used for further clinical or computational post-hoc analysis such as nucleus segmentation. We observe that the high-risk regions exhibit a very high concentration and different arrangement of epithelial cells (red) which is commonly associated with the origin of various cancer types [Coradini et al., 2011].

into epithelial cells, lymphocytes, macrophages, and neutrophils using a HoverNet [Graham et al., 2019] pre-trained on the MoNuSAC dataset [Verma et al., 2021]. For the high-risk regions, we observe a higher concentration of epithelial cells that are commonly associated with various cancer origins [Coradini et al., 2011], which we see as initial validation that the high-risk patches identified by the model capture salient biological signal.

# 7 Conclusion

We introduce HEALNet, a flexible *hybrid early-fusion* approach for multimodal learning. HEALNet has several distinctive and beneficial properties, suitable for applications in biomedical domains: 1) it preserves structural signal of each modality through modality-specific attention, 2) it learns cross-modal interactions due to its iterative architecture, 3) it effectively handles missing modalities, and 4) it enables easy model inspection. The experimental evaluation highlights the importance of fusing data early in the model pipeline to capture the cross-modal signal leading to better overall model performance. While in this work we focus only on survival analysis using modalities from digital pathology and genomic data, we believe that our framework can also be extended to other domains (and modalities) such as radiology or precision oncology, as well as other tasks such as diagnosis or predicting treatment response.

## Broader Impact Statement

HEALNet is a novel and flexible multimodal approach able to leverage complex heterogeneous biomedical data. Technological advances in medical data collection, such as high-resolution histopathology and high-throughput genomic sequencing, have inspired the development of novel multimodal approaches, able to address a variety of challenging modelling tasks in many biomedical applications. Our work extends the state-of-the-art in this area, introducing a performant architecture with several distinctive and beneficial properties suitable for many clinically-relevant applications. Specifically, we demonstrate the utility of our architecture on four real-world applications related to cancer prognosis using histopathology and multi-omic data collected by The Cancer Genome Atlas (TCGA) consortium.

As such, the primary aim and impact of our work is advancing data analysis capabilities in critical domains such as medicine and biology, focusing on complex tasks that require simultaneous modelling of multimodal data. The ability of our architecture to effectively handle missing modalities during inference can support many clinically relevant scenarios. For instance, this includes scenarios where HEALNet has been trained using costly data, but it can still be used to predict outcomes for patients in clinics that do not have access to sophisticated data collection technologies. Moreover, as HEALNet is explainable 'by design', it can enable easy model inspection as well as insights into the cross-modal interactions, which can facilitate trustworthiness and better adoption in such critical domains.

To this end, our work has only been evaluated in a strictly research setting. Further applications of our work in scenarios with sensitive data, such as clinical practice, introduce some challenges. As our primary focus is biomedical applications, data privacy must be carefully managed. Furthermore, as HEALNet is intended to serve as a decision-aiding tool, it bears risks of decision bias. Mitigating this involves further extensive evaluations, clinical trials, and medical regulation to ensure its reliability and safety before wider adoption in clinical settings.

## Acknowledgments and Disclosure of Funding

The authors would like to thank Philip Schouten for his insightful clinical feedback. KH acknowledges support from the Gates Cambridge Trust via the Gates Cambridge Scholarship. NS and MJ acknowledge the support of the U.S. Army Medical Research and Development Command of the Department of Defense; through the FY22 Breast Cancer Research Program of the Congressionally Directed Medical Research Programs, Clinical Research Extension Award GRANT13769713. Opinions, interpretations, conclusions, and recommendations are those of the authors and are not necessarily endorsed by the Department of Defense.

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

# Appendix

## HEALNet: Multimodal Fusion for Heterogeneous Biomedical Data

## A   HEALNet Pseudocode

---

**Algorithm 1** HEALNet

---

**Input**: Training data $X_m = \{x_1^{(1)}, \ldots, x_j^{(n)}\}$; for modalities $m \in \{1, \ldots, j\}$ and samples $N \in \{1, \ldots, n\}$, where $m, N \in \mathbb{N}$.

**Input**: Survival training labels $Y = \{y^{(1)}, \ldots, y^{(n)}\}$ for $S \in \{1, \ldots, s\}$ survival bins, where $S, y^{(i)} \in \mathbb{N} \, \forall i = 1, \ldots, n$.

**Input**: Number of fusion layers $d \in \mathbb{N}$.

**Output**: Logits of survival predictions $\hat{Y}_{logits} \in \{\hat{y}^{(1)}, \ldots, \hat{y}^{(n)}\}$, where $\hat{y}^{(i)} \in \mathbb{R}, \forall i = 1, \ldots, n$.

  1: $S_0 \leftarrow \mathbb{R}^{c_l \times d_l}$ // $S_0$ is the latent array with $S_0[i,j] \sim \mathcal{U}(0,1) \, \forall i \in \{1, \ldots, c_l\}, j \in \{1, \ldots, d_l\}$ for latent channels $c_l$ and latent dimensions $d_l$.

  2: **for** fusion layer $l = 0, \ldots, d-1$ **do**

  3:      $S_{l+j} \leftarrow FusionUpdate(S_l, X_m)$ // Equation 3

  4: **end for**

  5: $\hat{Y}_{logits} \leftarrow LinearHead(S_T)$ // where total timesteps $T = md \in \mathbb{N}$

  6: $\hat{Y}_{hazard} = sigmoid(\hat{Y}_{logits})$

  7: $Loss = NegativeLogLikelihood(Y, \hat{Y}_{hazard})$

  8: **return** $\hat{Y}_{hazard}$

---

**Algorithm 2** Hybrid Early-fusion Layer (*FusionUpdate*)

---

**Input**: Latent array $S_t \in \mathbb{R}^{c_l \times d_l}$ at time step $t \in \{1, \ldots, T\}$.

**Input**: Training data $X_m = \{x_1^{(1)}, \ldots, x_j^{(n)}\}$ for modalities $m \in \{1, \ldots, j\}$ and samples $N \in \{1, \ldots, n\}$, where each $x_i^{(m)} \in \mathbb{R}^{d_x}$ and $m, N, d_x \in \mathbb{N}$. $d_x$ corresponds to each modality's channel dimensions.

**Output**: $S_{t+j} \in \mathbb{R}^{c_l \times d_l}$.

  1: **for** modality $m = 1, \ldots, j$ **do**

  2:      $Q_m \leftarrow W_m^{(q)} S_{t+m-1}$ where $W_m^{(q)} \in \mathbb{R}^{d_l}$ // query

  3:      $K_m \leftarrow W_m^{(k)} X_m$ where $W_m^{(k)} \in \mathbb{R}^{d_x}$ // key

  4:      $V_m \leftarrow W_m^{(v)} X_m$ where $W_m^{(v)} \in \mathbb{R}^{d_x}$ // value

  5:      $\phi^{a_m} \leftarrow \{W_m^{(q)}, W_m^{(k)}, W_m^{(v)}\}$

  6:      $a_m^{(t)} = \alpha(X_m, S_t; \phi^{a_m})$ // Attention, Equation 1

  7:      $S_{t+m} = \psi(S_t, a_m^{(t)})$ // Latent Update, Equation 2

  8:      $S_{t+m} \leftarrow SNN(S_{t+m})$

  9: **end for**

10: **return** $S_{t+j}$

---

# B Experimental Results

While the main paper includes selected experimental results, where we only show the best uni-modal performance, the full list of experimental results can be found below.

Table 3: Full results of mean and standard deviation of the concordance Index on four survival risk categories. We trained HEALNet and all baselines on four The Cancer Genome Atlas (TCGA) tasks and report the performance on the hold-out test set across five cross-validation folds. A selection of these results are presented in Table 1.

| Modalities | Model (+Fusion Type) | BLCA | BRCA | KIRP | UCEC |
|---|---|---|---|---|---|
| Omic | Porpoise (Late) | $0.590 \pm 0.043$ | $0.580 \pm 0.027$ | $0.780 \pm 0.035$ | $0.550 \pm 0.026$ |
| | MCAT (Intermediate) | $0.552 \pm 0.036$ | $0.484 \pm 0.045$ | $0.721 \pm 0.084$ | $0.542 \pm 0.065$ |
| | MultiModN (Sequential) | $0.575 \pm 0.037$ | $0.422 \pm 0.070$ | $0.764 \pm 0.149$ | $0.440 \pm 0.086$ |
| | HEALNet (ours) | $0.606 \pm 0.019$ | $0.556 \pm 0.035$ | $0.771 \pm 0.135$ | $0.509 \pm 0.016$ |
| WSI | Porpoise (Early) | $0.540 \pm 0.030$ | $0.550 \pm 0.037$ | $0.520 \pm 0.037$ | $\mathbf{0.630 \pm 0.028}$ |
| | MCAT (Intermediate) | $0.556 \pm 0.039$ | $0.489 \pm 0.039$ | $0.533 \pm 0.099$ | $0.602 \pm 0.068$ |
| | MultiModN (Sequential) | $0.500 \pm 0.040$ | $0.510 \pm 0.055$ | $0.484 \pm 0.102$ | $0.489 \pm 0.022$ |
| | HEALNet (ours) | $0.487 \pm 0.046$ | $0.529 \pm 0.042$ | $0.518 \pm 0.123$ | $0.558 \pm 0.072$ |
| Omic + WSI | Porpoise (Late) | $0.620 \pm 0.048$ | $0.630 \pm 0.040$ | $0.790 \pm 0.041$ | $0.590 \pm 0.034$ |
| | MCAT (Intermediate) | $0.620 \pm 0.040$ | $0.589 \pm 0.073$ | $0.789 \pm 0.087$ | $0.589 \pm 0.062$ |
| | MultiModN (Sequential) | $0.551 \pm 0.060$ | $0.582 \pm 0.084$ | $0.753 \pm 0.152$ | $0.610 \pm 0.121$ |
| | Perceiver (Early) | $0.565 \pm 0.042$ | $0.566 \pm 0.068$ | $0.783 \pm 0.135$ | $0.623 \pm 0.107$ |
| | HEALNet (ours) | $\mathbf{0.668 \pm 0.036}$ | $\mathbf{0.638 \pm 0.073}$ | $\mathbf{0.812 \pm 0.055}$ | $\mathit{0.626 \pm 0.107}$ |

**Additional experiments**

We provide an extended analysis of the performance of HEALNet on combining tabular and time series modalities, evaluated on intensive care data from the Medical Information Mart for Intensive Care (MIMIC-III) Johnson et al. [2016]. We train models for two separate tasks: predicting patient mortality ('MORT'), formulated as a multi-class classification task; and disease classification ('ICD-9' codes), which we formulate as a binary classification task. We use both clinical variables and small time series data on various vital signs measured at 24 time steps. Both tasks have sample size of $n = 32616$ and the same feature set for different task labels. We report the average AUC in the case of ICD9 and Macro-AUC ("one-vs-rest") in the case of MORT, averaged across five folds.

Table 4: Mean and standard deviation of classification performance. We trained HEALNet and all baselines on two MIMIC-III tasks and report the performance on the test set across five folds. HEALNet outperforms all multi-modal and uni-modal baselines in classification performance.

| Model (+Fusion Type) | ICD9 | MORT |
|---|---|---|
| UniModal (Tabular) | $0.731_{\pm 0.023}$ | $0.658_{\pm 0.000}$ |
| UniModal (Time Series) | $0.700_{\pm 0.013}$ | $0.715_{\pm 0.016}$ |
| Perceiver (Early) | $\mathit{0.733_{\pm 0.028}}$ | $\mathit{0.723_{\pm 0.015}}$ |
| MultiModN (Sequential) | $0.500_{\pm 0.000}$ | $0.500_{\pm 0.000}$ |
| MCAT (Intermediate) | $0.500_{\pm 0.000}$ | $0.500_{\pm 0.000}$ |
| HEALNet (ours) | $\mathbf{0.767_{\pm 0.022}}$ | $\mathbf{0.748_{\pm 0.009}}$ |

# C    Dataset Details

The results shown in this paper here are based upon data generated by the TCGA Research Network: https://www.cancer.gov/tcga. The Cancer Genome Atlas (TCGA) is an open-source genomics program run by the United State National Cancer Institute (NCI) and National Human Genome Research Institute, containing a total of 2.5 petabyts of genomic, epigenomic, transcriptomic, and proteomic data. Over the years, this has been complemented by multiple other data sources such as the whole slide tissue images, which we use in this project. It contains data on 33 different cancer types for over 20,000 patients. Across the four cancer sites in the scope of this paper, we process a total of 2.5 Terabytes of imaging and omics data, the vast majority of which is taken up by the high-resolution whole slide images. Specifically, the four cancer sites are:

- Urothelial Bladder Carcinoma (BLCA): Most common type of bladder cancer, where the carcinoma starts in the urothelial cells lining the inside of the bladder.

- Breast Invasive Carcinoma (BRCA): Commonly referred to as invasive breast cancer refers to cancer cells that have spread beyond the ducts of lobules into surrounding breast tissue.

- Kidney Renal Papillary Cell Carcinoma (KIRP): Type of kidney cancer characterised by the growth of papillae within the tumour which is multi-focal, meaning that they frequently occur in more than one location in the kidney.

- Uterine Corpus Endometrial Carcinoma (UCEC): Most common type of uterine cancer arising in the endometrium, i.e., the lining of the uterus.

Table 5: Overview of data availability and dimensionality of the four TCGA datasets used for experiments.

| Property | BLCA | BRCA | KIRP | UCEC |
|---|---|---|---|---|
| Slide samples | 436 | 1,019 | 297 | 566 |
| Omic samples | 437 | 1,022 | 284 | 538 |
| Overlap (n used) | 436 | 1,019 | 284 | 538 |
| Omic features used | 2,191 | 2,922 | 1,587 | 1,421 |
| Sample WSI resolution (px) | 79,968 x 79,653 | 35,855 x 34,985 | 72,945 x 53,994 | 105,672 x 71,818 |
| Censorship share | 53.9% | 86.8% | 84.5% | 85.5% |
| Survival bin sizes | [72, 83, 109, 172] | [403, 289, 172, 155] | [43, 56, 113, 72] | [68, 143, 83, 244] |
| Disk space (GB) | 594 | 883 | 275 | 756 |

# D  Hyperparameters

Table 6: Overview of Hyperparameters used in the final code implementation. Parameters that are consistent across datasets can be set in the `config/main_gpu.yml` and dataset-specific parameters can be set in `config/best_hyperparameters.yml`

| Scope | Paramter | BLCA | BRCA | KIRP | UCEC |
|---|---|---|---|---|---|
| Shared | Output dims | 4 | 4 | 4 | 4 |
| | Patch Size | 256 | 256 | 256 | 256 |
| | Loss | NLL | NLL | NLL | NLL |
| | Loss weighting | inverse | inverse | inverse | inverse |
| | Subset | uncensored | uncensored | uncensored | uncensored |
| | Batch Size | 8 | 8 | 8 | 8 |
| | Epochs | 50 | 50 | 50 | 50 |
| | Early stopping patience | 5 | 5 | 5 | 5 |
| | Scheduler | OneCycle LR | OneCycle LR | OneCycle LR | OneCycle LR |
| | Max LR | 0.008 | 0.008 | 0.008 | 0.008 |
| | Momentum | 0.92 | 0.92 | 0.92 | 0.92 |
| | Optimizer | Adam | Adam | Adam | Adam |
| | L1 reg | 0.00001 | 0.000006 | 0.00004 | 0.0003 |
| HEALNet | Layers | 2 | 2 | 5 | 2 |
| | Shared Latent dims | 25 x 119 | 25 x 119 | 25 x 119 | 25 x 119 |
| | Attention heads | 8 | 8 | 8 | 8 |
| | Dims per head | 16 | 63 | 27 | 103 |
| | Attention dropout | 0.08 | 0.46 | 0.32 | 0.25 |
| | Feedforward dropout | 0.47 | 0.36 | 0.05 | 0.06 |
| MCAT/MOTCAT | Model Size Omic | 256x256 | 256x256 | 256x256 | 256x256 |
| | Model Size WSI | 1024 x 256 x 256 | 1024 x 256 x 256 | 1024 x 256 x 256 | 1024 x 256 x 256 |
| | Dropout | 0.1 | 0.25 | 0.25 | 0.25 |
| | Fusion Method | bilinear | bilinear | concat | concat |
| Perceiver | Layers | 2 | 2 | 5 | 2 |
| | Latent dims | 25 x 119 | 17 x 126 | 17 x 62 | 16 x 65 |
| | Attention heads | 8 | 8 | 8 | 8 |
| | Attention dropout | 0.08 | 0.46 | 0.32 | 0.25 |
| | Feedforward dropout | 0.47 | 0.36 | 0.05 | 0.06 |

# E   Sensitivity to the regularisation mechanism

We found that HEALNet can be sensitive to the choice of regularisation mechanisms, even though the current regularisation techniques, such as L1 + SNN, have shown to be effective.

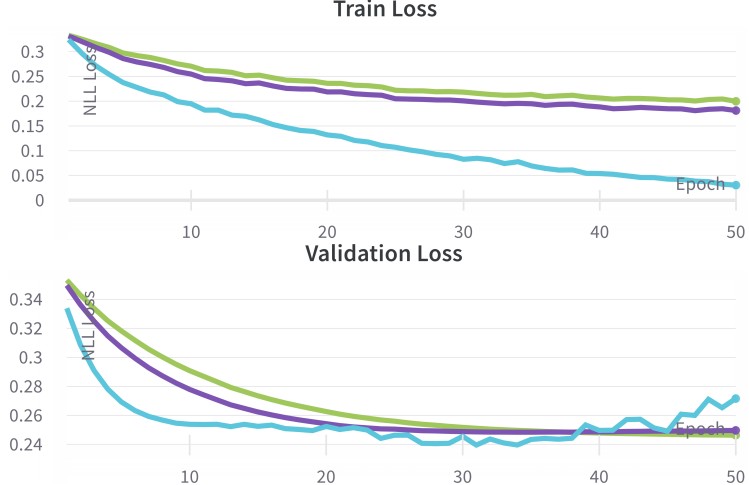

Figure 4: Effect of the regularisation mechanism. We show the train (top) and validation (bottom) losses on the KIRP dataset, of HEALNet variants with no regularisation (blue), only L1 regularisation (indigo), and L1 regularisation + a self-normalising network layer (green).

