# OpenReview forum: "HEALNet: Multimodal Fusion for Heterogeneous Biomedical Data"
_NeurIPS.cc/2024/Conference — NeurIPS 2024 poster_

### Official Review · Reviewer_YCoh · 2024-07-12

**Soundness:** 3
**Presentation:** 3
**Contribution:** 3
**Rating:** 5
**Confidence:** 4

**Summary:**

The paper proposes an attention based early-fusion method for multi-modal learning. Specifically the paper uses a perceiver style iterative updates to a latent vector using  a fusion layer. The fusion layer sequentially updates the latent vector using cross-attention with each of the modalities. The approach supports cross-modal interactions and provides a way to handle missing modalities at inference time.

The authors compare the approach against existing baselines for fusing biomedical modalities and  show promising results on the survival prediction task.

**Strengths:**

The paper is well written and provides good context on existing methods.

The proposed approach is simple and explained clearly.

The iterative style fusion is useful as it can support cross-modal interactions without exploding the embedding dimension and can also handle missing modalities at inference time.

**Weaknesses:**

Its unclear how much the iterative fusion is helpful as the parameters are shared. Can the authors provide more some information about when the iterative fusion is useful and how many iterations are typically needed.

The proposed iterative style updates to the latent vector seems similar to an LSTM style aggregation. In many VLM and multimodal papers (https://arxiv.org/pdf/2405.09818, https://arxiv.org/pdf/2304.08485), the modality specific tokens are projected to a shared latent space and are aggregated through self-attention across the shared space. This provides more opportunity for cross modal interactions than iterative cross-attention. Have the authors compared with such approaches?

The results are shown for a specific task (survival prediction) and specific modalities where the datasets are small and models are prone to overfitting. Its unclear if this approach works well for problems which need more complex interactions and where dataset sizes are larger.

**Questions:**

Its unclear how the features from different patches in the WSI aggregated? What is the embedding dimension?

Do the early-fusion models have more parameters? If so are they more prone to overfitting on these datasets?

Since the latent array is updated sequentially with information from different modes, how much does the order of fusing modalities impact the result?

**Limitations:**

Given the weaknesses and open questions, I'm basing my decision but would be open to changing the decision post rebuttal once the questions and weaknesses are addressed.

---

> ### Author Rebuttal · Authors · 2024-08-06
>
> ### **Comments on weaknesses:**
>
> **Usefulness of iterative fusion:**
>
> In short, having multiple fusion layers (i.e., iterations) helps to prevent overfitting. During development, we found that the modality-specific updates only learn how to operate on the latent embedding $S$ directly after the previous modality update if you only have one fusion layer. Increasing the number of fusion layers means that $S$ is updated with all modalities multiple times such that information from each modality becomes part of the context vector for every other modality. We share the weights between the fusion layers to prevent layer specificity such that the modality-specific updates become context-agnostic. This is why the model also delivers robust performance when skipping a modality at inference time (Table 2).
>
> Regarding the number of fusion layers needed, we ran some initial ablation studies during development, where we did not find a clear pattern on the exact number of fusion layers needed, but found that $d>1$ was an effective regulariser.
>
> **Cross-modal interactions & other architectures:**
>
> Using the shared latent space projection in a “monolithic” architecture is something that we originally considered and decided against for three main reasons:
>
> - **Don’t align what shouldn’t be aligned:** First, we would like to point out a crucial distinction of multimodal modelling in the Vision & Language (V&L) domain and on biomedical modalities beyond V&L.
>     - **Vision & Language** models rely on the assumption that the same semantic concept is explicitly expressed across modalities (e.g., an image and its description). In other words, both modalities follow the same training distribution and V&L models consequently align the latent representations to reflect this distribution across both modalities, such as through the contrastive objectives in the linked Chameleon paper.
>     - For **Biomedical modalities beyond V&L**, we don’t want to make this assumption. Often, the signal from different modalities (e.g., morphological features, genomics, and transcriptomics) follows different distributions and we want to maintain this separation to build a more predictive model. If we used projections into the same latent space combined with contrastive objectives, this would align the signal from the separate distributions. This defeats the purpose of cross-modal learning for these modalities as it becomes more difficult to learn from both distributions.
>     - **We have benchmarked** several approaches which leverage projection and alignment in the same latent space. The MCAT baseline also aligns both modalities directly through the “genomics-guided cross-attention” where the dot product of the WSIs and the multiomic data is minimised. Another example is the Perceiver (Early Fusion) baseline which is first combining the input modalities to a common tensor representation and then passes them through multiple cross-and self-attention layers. HEALNet **consistently outperforms** these methods (Table 1).
> - **Scaling up to many modalities:** The iterative attention mechanism alleviates the use of fusion operators that can result in high-dimensional latents (e.g., Kronecker product [1]). This allows for combining many  modalities while preserving the structural information of each as well as learning cross-modal interactions.
> - **Improved handling of missing modalities:** Another benefit of an iterative architecture instead of a monolithic one is that each layer pass does not require all modalities at the same time.
>
> **Larger datasets:**
>
> We would like to point out that for cancer pathology tasks The Cancer Genome Atlas (TCGA) contains by far the largest patient cohorts available in the public domain with each cancer cohort being over 500GB in size (L192-198).
>
> To show HEALNet’s performance on larger sample sizes, we ran some additional experiments comparing HEALNet to a subset of baselines (due to resource constraints in the rebuttal period) on a disease classification (ICD9) and patient mortality (MORT) prediction tasks of MIMIC-III (n=32,616). The modalities in MIMIC are tabular EHR features and time series on vital signs and test results. The results are shown in the general comment of the rebuttal.
>
> ### **Responses to questions**
>
> - **[…]unclear how the features from different patches in the WSI aggregated? […]**
>     - The final embedding dimension is a 2D Vector of size `num_patches x encoding_dim` . The `encoding_dim` refers to the dimensions of the Kather100k pre-trained encoder, which results in a 2048-dimensional feature vector per patch (see L195).
> - **Do the early-fusion models have more parameters? […]**
>     - This highly depends on the early fusion method used. A commonly used early fusion methods is the Kronecker product [1] which is constructing a high-dimensional input tensor, meaning that models using Tensor fusion would require a higher number of parameters in the earlier layers than HEALNet. However, other early fusion models use aggregation methods (e.g., concatenation and averaging across excess channels, as done in the Perceiver). In these cases, the parameters required in the early layers would be lower than HEALNet’s, but notably this approach also collapses an entire dimension, which may remove salient signal.
> - **[…] how much does the order of fusing modalities impact the result?**
>     - During development of HEALNet, we performed preliminary analyses on the influence of the order of modalities on the downstream performance: We did not find any significant difference. Since all modalities become the context of each other’s update as long as $d>1$, the order of the modalities does not matter.
>
> References:
>
> [1] Zadeh, A. et al (2017). Tensor fusion network for multimodal sentiment analysis.

---

> > ### Comment · Reviewer_YCoh · 2024-08-12
> >
> > >having multiple fusion layers (i.e., iterations) helps to prevent overfitting
> >
> > Thanks I think it makes sense that this helps reduce influence of the order of modalities. The choice of weight sharing makes sense with this design, I think its worth highlighting this in the paper.
> >
> > >Don’t align what shouldn’t be aligned
> >
> > While I do agree the VL models are aligned, the idea of projecting to a shared representation space in which interactions happen is also being used in the cross-attention interaction, although the iterative fusion does reduce dependency slightly.
> > If the modalities are complementary its unclear why a concatenation approach doesn't work as well, while it may not handle missing modalities.
> >
> > Not sure I understand the perceiver early fusion setup well. Are the tokens from different modalities concatenated and passed through iterative fusion? Wouldn't it have similar paramters to HEALNet then (the Q,K,V projection dims remain similar)
> >
> > > The final embedding dimension is a 2D Vector of size num_patches x encoding_dim
> > This is indeed a large sequence length.

---

> ### Author Response · Authors · 2024-08-12
>
> Thank you very much for responding to our rebuttal! Per your suggestion, we will further highlight the role of having multiple fusion layers in the manuscript. We briefly wanted to clarify the open points below:
>
> > “If the modalities are complementary its unclear why a concatenation approach doesn't work as well, while it may not handle missing modalities.”
>
> We agree that, in principle, the concatenation approach would work well if all modalities had the same tensor dimensions. However, if we are processing structurally different modalities (e.g., 1D vs 2D vs 3D tensors), one would need to flatten or aggregate the excessive dimensions to ensure shape consistency for the projection. This leads to two problems: 1) we may end up with excessively large 1D tensors which 2) remove spatial signal if we e.g., flatten an image representation. In practice, we cannot always assume shape consistency between our modalities and the presence of all modalities for every sample (especially in clinical practice) - both of these aspects informed our design choices. We touch on these aspects in L89 & L111-114.
>
> > “Are the tokens from different modalities concatenated and passed through iterative fusion? Wouldn't it have similar paramters to HEALNet then (the Q,K,V projection dims remain similar)”
>
> Contrary to the Perceiver, in HEALNet each modality has its *own* Q,K,V projection (as indicated in Figure 1), which allows us to keep the input data intact irrespective of the tensor dimensions. That is, HEALNet does not require to aggregate excessive dimensions which helps to maintain the structural signal of each modality.

---

> > ### Comment · Reviewer_YCoh · 2024-08-13
> >
> > Thanks to the authors for addressing my questions.
> >
> > The method provides a simple way to aggregate information from different modalities without losing their structural information. However its unclear how the results show here translate to other problems.
> >
> > Based on this I am updating my score from 4 to 5.

---

### Official Review · Reviewer_5LQ9 · 2024-07-12

**Soundness:** 3
**Presentation:** 2
**Contribution:** 3
**Rating:** 5
**Confidence:** 3

**Summary:**

The authors present HEALNet as an early fusion approach for integrating different data modalities. HEALNet utilizes an end-to-end training process with an additive method for combining modalities, rather than handling them in parallel. This strategy enables HEALNet to scale and adapt effectively to datasets of varying sizes and characteristics without requiring explicit pre-training.

**Strengths:**

- The idea of the authors to implement an additive approach to combine modalities instread of processing them in parallel is promising.

**Weaknesses:**

- The methodology is not clearly written and is difficult to understand.
- The presentation also falls short. For instance, Figure 3 features a bar plot that is unreadable.
- Additionally, the color scheme in the attention map seems unusual; typically, a blue-red gradient is used as it is more intuitive.
- The visualization analysis is lacking. There is only one figure displaying the model's inspection capabilities, which features a WSI attention map.
- The iterative attention mechanism, with multiple steps and updates, can be computationally expensive. This could be particularly challenging when dealing with very high-dimensional data or when scaling up to larger datasets.
- There is no ablation study regarding the number of the fusion layers. The number of fusion layers is a critical parameter that can significantly affect the computational efficiency of the model.

**Questions:**

- Could you include metrics such as as Floating Point Operations (FLOPs) and the number of trainable parameters to provide a more comprehensive understanding of the computational cost and efficiency of the HEALNet approach?
- We have seen in recent papers [Transcriptomics-guided Slide Representation Learning in Computational Pathology](https://arxiv.org/html/2405.11618v1) that in cases where representations come from different sources or modalities the different  repressentations are aligned in a meaningful way to facilitate effective learning and integration. How does your model handle discrepancies in feature scales or semantic differences between modalities?

**Limitations:**

The methodology needs to be clearer. The dimensions of the cross-attention layers are not clearly defined, which complicates understanding of how these layers align and integrate features from modalities of differing dimensions. Additionally, the visualizations and illustrations require significant improvement, as current figures, such as bar plots and attention maps, are difficult to interpret and hard to read. Enhancing these visual elements and providing clearer explanations of the methodology will improve the quality of the  paper.

---

> ### Author Rebuttal · Authors · 2024-08-06
>
> ### **Comments on weaknesses**
>
> - **“methodology is not clearly written and difficult to understand”**: We would like to point to the step-by-step methodology, which is discussed in detail in Section 3, illustrated in Figure 1, and presented as pseudocode in Appendix A. We believe this to be a very detailed description of HEALNet. Nevertheless,  we would be grateful if the reviewer could specify which parts of the methodology are unclear, so we can further improve the clarity of our manuscript.
> - **“[…] Figure 3 features a bar plot that is unreadable”**: We would kindly like to point out that the aim of Figure 3 is to provide a high-level *illustration* of the general inspection capabilities of HEALNet that help users validate the model, as well as understand the results. We believe we mention this in the caption of Figure 3. In this work, Figure 3 is meant to showcase the capabilities of HEALNet, rather than to deep dive into the biology of UCEC. The size/formatting of the Figure 3 is due to page limitations. Still, Figure 3 is sufficiently high resolution that the individual feature names can be read on the PDF version of the paper. Nevertheless, we appreciate the feedback and will provide a larger version of the image in the Appendix.
> - **“iterative attention mechanims […] can be computationally expensive”**: The computational cost of the iterative attention mechanism is alleviated by sharing weights across the fusion layers, which makes the number of parameters independent of the fusion layers. It is worth noting that the existing tasks of the paper are already dealing with very high-dimensional datasets (see L181 onwards). To show that HEALNet also works for datasets with a larger sample size, we ran some additional experiments comparing HEALNet to relevant baselines on a disease classification (ICD9) and patient mortality (MORT) prediction tasks of MIMIC-III (n=32,616). The table below shows the AUC and Macro AUC for the respective tasks. The modalities in MIMIC are tabular electronic health record features and time series on vital signs and test results.
>
> | Model/Dataset | ICD9 (n=32,616) [AUC] | MORT (n=32,616)  [MacroAUC] |
> | --- | --- | --- |
> | Uni-modal (EHR) | 0.731 ± 0.023 | 0.658 ± 0.001 |
> | Uni-modal (Time Series) | 0.700 ± 0.013 | 0.715 ± 0.016 |
> | Perceiver (Early) | 0.733 ± 0.028 | 0.723 ± 0.015 |
> | Porpoise (Late) | 0.628 ± 0.020 | 0.617 ± 0.015 |
> | HEALNet | 0.767 ± 0.022 | 0.748 ± 0.009 |
> - **“[...] number of fusion layers is a critical parameter that can significantly affect the computational efficiency of the model”:** As previously mentioned, the number of fusion layers does not affect the computational efficiency of the model due to the implementation of weight sharing.
>
> ### **Responses to questions**
>
> - **Handling discrepancies in feature scales and semantic differences between modalities:** We would like to point out a crucial distinction between multimodal modelling tasks in the “standard” Vision & Language (V&L) setting and ones over Biomedical modalities (beyond V&L):
>     - Current **Vision & Language** approaches rely on the assumption that the same semantic concepts are expressed within the modalities (e.g., an image and its text description). Moreover, these semantic concepts are typically explicit (e.g., an image with a bird and a description that includes ‘bird’). Therefore, both modalities follow the same training distribution. In turn, V&L approaches align the latent representations to reflect that distribution across both modalities, and leverage these explicitly expressed semantic concepts through a contrastive objectives, such as the one in the linked Tangled paper [1].
>     - For **Biomedical modalities beyond V&L**, we generally cannot make such assumptions. Often times, the signal from different modalities (e.g., morphological features, genomics, and transcriptomics) follows different distributions and we want to maintain that separation to build a more predictive model. If we used projections into the same latent space combined with contrastive objectives, this would likely mute the signal from the separate modalities. This defeats the purpose of cross-modal learning for these modalities as it becomes more difficult to learn from both distributions. It is worth noting that while we separately align the latent representation to each modality, we never align the modality representation with each other to prevent such a muting effect. Moreover, on a broader note, if we were to make a similar assumption as in [1], such an approach would be applicable and scale only to two modalities - which is the case in [1], where the authors learn from WSI and gene expression profiles only. HEALNet, on the other hand, is robust and can readily scale to many different modalities of any size, as shown in the MIMIC experiments above.
> - While we appreciate and thank the reviewer for the reference [1],  we would like to point out that paper [1] was published just three days before the NeurIPS deadline. Nevertheless, we have benchmarked some approaches that perform a projection and alignment in the same latent space. Specifically, the MCAT [2] baseline also aligns both modalities directly through the “genomics-guided cross-attention” where the dot product of the WSIs and the multiomic data is minimised. Another example is the Perceiver (Early Fusion) [3] baseline which is first combining the input modalities to a common tensor representation and then passes them through multiple cross-and self-attention layers. Note that HEALNet consistently outperforms both of these methods, as shown in Table 1 and Section 5.
>
>
>
> [1] Jaume et al. (2024) Transcriptomics-guided Slide Representation Learning in Computational Pathology
>
> [2] Chen, R.J. et al. (2021). Multimodal co-attention transformer for survival prediction in gigapixel whole slide images.
>
> [3] Jaegle, A., et al. (2021) Perceiver: General perception with iterative attention.

---

> > ### Comment · Reviewer_5LQ9 · 2024-08-13
> >
> > I have reviewed the authors' rebuttal and appreciate the clarifying comments. However, my question regarding the inclusion of metrics such as Floating Point Operations (FLOPs) and the number of trainable parameters remains unanswered. So, does my question about the number of iterations that are required. Therefore, I intend to maintain my score.

---

> ### Author Response · Authors · 2024-08-13
>
> Thank you for responding to our rebuttal. In our initial response we briefly discuss the computational complexity of HEALNet, and specifically the role of the fusion layers (as per the last weakness point). We also added new experiments to show that HEALNet is able to scale seamlessly to larger datasets, where the computational cost of the iterative attention mechanism is alleviated by sharing weights across the fusion layers. This makes the number of parameters independent of the number of fusion layers (i.e., iterations).
>
> More formally, for any multimodal model, it is important to scale with both number of samples $n$ and the number of modalities $m$. For instance, MCAT (our most competitive baseline) is natively designed for two modalities. To scale this approach to $m>2$, one would need to calculate the modality-guided cross-attention for all unique pairwise combinations ${m \choose 2} = \frac{m(m-1)}{2}$ which is $\mathcal{O}(m^2)$.
>
> A **key advantage of HEALNet’s** sequential setup is that it scales linearly. Since the cross-attention and SNN layers used within the fusion layers scale with $\mathcal{O}(n)$, each fusion layer has a complexity of $\mathcal{O}(mn)$ such that the runtime then mainly depends on the number of fusion layers $d$. Note that, HEALNet we consider $d$ to be a hyper-parameter with the values provided in Appendix E. We plan to post the number of parameters of HEALNet and the baselines on the KIRP dataset later today.
>
> We acknowledge that the actual runtime will depend on several other factors such as the size of the modality-specific embeddings within each model. As we didn't measure FLOPs at this instance, we provide the BigO analyses of our method with respect to the other baselines. Assuming we use the same encoders for each model, the time complexity for each fusion operation is summarized below.
>
> | Model | Time Complexity with number of modalities $m$ and samples $n$ |
> | --- | --- |
> | Perceiver (early, concat) | $\mathcal{O}(nm)$  |
> | MultiModN (sequential) | $\mathcal{O}(nm)$ |
> | MCAT(intermediate) | $\mathcal{O}(m^2n)$ |
> | Porpoise (late, Kronecker) | $\mathcal{O}(m^2n)$ |
> | HEALNet | $\mathcal{O}(nm)$ |
>
> We hope that this clarified the remaining concerns and will be reflected in your updated score.

---

> > ### Comment · Reviewer_5LQ9 · 2024-08-13
> >
> > I have carefully considered the authors' rebuttal and appreciate the additional clarity provided. As a result, I am updating my score to 5.

---

> > > ### Author Response · Authors · 2024-08-13
> > >
> > > Thanks for acknowledging our response and your positive outlook - we are glad that we were able to further clarify our work. We would appreciate if you could update your increased positive score, as stated in your last comment, in the system before the deadline of the discussion period.

---

### Official Review · Reviewer_ZYCi · 2024-07-12

**Soundness:** 4
**Presentation:** 4
**Contribution:** 3
**Rating:** 7
**Confidence:** 4

**Summary:**

This paper presents HEALNet, a method for end-to-end multimodal fusion of mixed-type biomedical data. In contrast to methods like feature concatenation or Kronecker product fusion, HEALNet employs an iterative cross-attention structure that operates on the raw input modalities, representing a hybrid between early and intermediate-fusion approaches. This allows for the necessary cross-modal interaction, while natively handling missing modalities at test time and aiding interpretability (by operating on raw data rather than embeddings). HEALNet outperforms existing state-of-the-art multimodal fusion methods on survival analysis from histopathology and multi-omic data.

**Strengths:**

- The problem motivation is very clear, and the explanation of related prior work is thorough. Care is taken to properly inform the reader of relevant background so that the authors can demonstrate why the proposed method is unique.
- The structure and logical flow of the paper is excellent. Tables and illustrations are of high quality throughout.
- The proposed method is interesting and is designed to solve domain-specific problems in biomedical multimodal fusion such as handling missing modalities and interpretability, while still enabling powerful cross-modal learning.
- The experiments appear to be soundly conducted, and results are convincing against competitive relevant baselines.

**Weaknesses:**

- Treatment of hyperparameters is somewhat confusing. It is unclear if Bayesian hyperparameter optimization (HPO) was applied to all methods or just to HEALNet. If HPO was applied to all methods, then why do their hyperparameters not appear in Table 5? These are important details to ensure a fair comparison between methods.
- The latent update in Equation 2 is unclear, and this seems to be a critical component of the proposed method. What *exactly* is this function $\psi(\cdot)$? Also, there is an additional parameter $\rho$ to $\psi(\cdot)$ in the pseudocode that is not present in Equation 2. This needs to be clarified.

**Questions:**

- Was Bayesian HPO applied to all methods or only HEALNet? Why does only HEALNet appear in Table 5? Please clarify this and which exact hyperparameters underwent optimization; for instance, it is confusing that all shared hyperparameters appear to be identical except the L1 regularization term.
- What exactly is the update function $\psi(\cdot)$? What is the parameter $\rho$ in the pseudocode, and why is it not present in Equation 2?

Minor comments:
- L18: Unnecessary to use MMML acronym if never used again
- L83: “additive approach to combining modalities (rather than handling them in parallel).” It is unclear what is meant by “additive” – sequential?
- L118: “fusing operators” -> “fusion operators”
- L125: Remove semicolon
- L145: I would write “…Learning Network and its key component, (b) the early fusion layer (as given in Equation 3).”
- Eq 4: Use \left[ and \right] to make square brackets larger
- L186: Can omit “contains”
- L187: Consider citing https://ieeexplore.ieee.org/abstract/document/10230356/, which addresses multimodal fusion methods to prevent overfitting in multimodal fusion. This is particularly relevant since this paper also considered fusion of histopathology imaging with tabular data, and forms results in contrast to the parameter-intensive Kronecker product as well.

**Limitations:**

Several limitations are adequately addressed in Section 6.

---

> ### Author Rebuttal · Authors · 2024-08-06
>
> ## **Responses to Weaknesses and Questions**
>
> **Hyperparameter optimization:** We would like to clarify that we ran the Bayesian Hyperparameter optimisation for all baselines. The shared parameters in Table 5 were ran equally for all baselines from Table 1. We acknowledge that Table 5 currently does not show the final hyperparameters of the baselines, but we will add these for the final manuscript. Please see the relevant hyperparameters that we tuned for the baselines below. Note that this choice was mostly informed by the original implementations of the respective papers.
>
> - MCAT/MOTCAT/Porpoise:
>     - `model_size_omic` : Embedding size of early layers for the omic data (pre-set choices between ‘large’ and ‘small’ from original papers)
>     - `model_size_wsi` : Same as model_size_omic, but for the WSI embedding
>     - `dropout` : Dropout applied after each layer of original implementation
>     - `fusion_method` : choice between concatenation and bilinear fusion
> - Perceiver (early):
>     - `layers` : total number of iterations
>     - `latent_dims` : dimensions of latent bottleneck
>     - `attention_heads` : number of attention heads
>     - `attn_dropout` : dropout after each self-attention layer
>     - `ff_dropout` : dropout after each feedforward pass
> - MultiModN:
>     - `embedding_dims` : size of latent embedding
>
>
> **Latent Update in Eq. 2:**
>
> - **Clarification on $\psi(\cdot)$**: In the main body, the update function $\psi(\cdot)$ represents the learnable function (i.e., the cross-attention update) instead of a fixed closed-form expression. In the pseudocode in Algorithm 2, given the attention matrix $attn \in \mathbb{R}^{b \times i \times j}$ for batch size $b$, number of queries $i$, and keys/values $j$, for modality $m$ at time step $t$ and the value matrix $v \in \mathbb{R}^{b \times j \times d}$ for the dimensions of the modality value vector $d$, the update mechanism is $\psi(S_t, a^{(t)}_m) = \sum_j attn_{b,i,j} \cdot v_{b,j,d}$.  We will make this part clearer in the updated version.
> - **Clarification on $\rho$:** Thanks for catching this. It is a leftover notation (typo) that we missed in the last version of the manuscript. We will correct this accordingly.

---

> > ### Comment · Reviewer_ZYCi · 2024-08-07
> >
> > I acknowledge that I have read the authors' rebuttal and thank them for the clarifying comments. I would encourage the authors to be as specific as possible when justifying hyperparameter choices, even if this material is relegated to the appendix. I will maintain my original score.

---

### Official Review · Reviewer_a9ao · 2024-07-12

**Soundness:** 2
**Presentation:** 3
**Contribution:** 2
**Rating:** 4
**Confidence:** 4

**Summary:**

The authors present a multi-modal fusion architecture, named HEALNet, which aims to preserve modality-specific structural information and capture cross-modal interactions in a shared latent space. HEALNet enables intuitive model inspection by learning directly from raw data inputs instead of opaque embeddings. The iterative paradigm can skip modality update steps to handle missing modalities. The authors validate the framework through multi-modal survival analysis on four cancer datasets.

**Strengths:**

1. The paper is well-organized and clearly written, with a logical flow of information that is easy to follow.
2. The iterative paradigm's ability to skip modality update steps to handle missing modalities is a practical approach.

**Weaknesses:**

1. HEALNet shares a very similar structure with Perceiver. The authors should highlight the innovative parts more clearly to help readers understand which aspects are novel and which are based on related work.
2. The evaluation is limited to survival analysis. Given that this is a multi-modal fusion framework, it would be beneficial to perform additional tasks, such as classification or regression, to better evaluate its performance.
3. The sample sizes for the four datasets used are relatively small. Instead of 5-fold repeated random subsampling, more repetitions should be conducted, and confidence intervals should be reported to ensure the reliability of the results.
4. In addition to the concordance index, other metrics or plots such as cumulative proportion surviving plots should be reported to better visualize the results.

**Questions:**

1. Higher performance was reported in the compared paper MOTCat [1]. Why did this happen?
2. Since the modalities are different, what is the rationale for sharing weights across all fusion layers in HEALNet? How is modality-specific structural information preserved during the fusion process?

**Limitations:**

1. The evaluation of HEALNet is limited to survival analysis, which may not fully capture its potential across different biomedical tasks.
2. The small sample sizes and limited number of datasets may affect the generalizability of the results.
3. The absence of a comprehensive set of performance metrics limits the understanding of the model's strengths and weaknesses.

---

> ### Author Rebuttal · Authors · 2024-08-06
>
> ## **Comments on Weaknesses**
>
> **Clarification on Novelty:** As you rightly point out, the general idea of iterative cross-attention with latent state passing has been previously suggested, which we acknowledge as a starting point for HEALNet’s architecture (L89, L113, L151). While other iterative modelling architectures (such as the Perceiver) handle a range of uni-modal tasks, HEALNet introduces a novel way of how iterative attention can be leveraged for learning effective multi-modal representations. Concretely, the Perceiver architecture requires concatenation of the input modalities. This becomes particularly problematic for modalities with misaligned dimensions and channels, as you either need to flatten all tensors or aggregate across channels, both of which trades off information for shape compatibility. The Perceiver paper demonstrates multimodal capabilities by concatenating audio and video embeddings, reducing the dimensions of the video and treats them as a single modality thereafter — this is identical to the Early Fusion baseline shown in Table~1, which HEALNet outperforms. HEALNet learns modality-specific latent updates to ensure that the modality shapes can be kept intact. Additionally, learning modality-specific updates becomes valuable in the handling of missing modalities. In such scenarios, an early fusion approach like the Perceiver would need to impute the missing values, which is noisy. In contrast, HEALNet’s setup allows to skip the missing updates at train and inference time.
>
> **Classification and Regression tasks:** We agree that the method is general which was a key design objective of the architecture. The focus on survival analysis in this paper was chosen for two main reasons.
>
> First, this paper focuses on multimodal data in the context of biomedical tasks. In these domains, most tasks that are a) clinically relevant and b) have sufficient data available to deal with time-to-event labels, where the outcomes of some study subjects is unobserved (right-censored data). Neither classification nor regression on these task labels is suited to handle time-to-event or the concept of censorship accurately.
>
> Second, it is worth noting that the survival analysis task was chosen to ensure comparability with relevant baselines. Under the hood, the survival analysis implementation is using a classifier to predict the hazard bins of a proportional hazards survival model.
>
> Architecturally, the only thing that would change to adapt to a classification task is the very final layer (”Head”) in Figure 1A.
>
> **Sample size**: In the field of cancer pathology, The Cancer Genome Atlas (TCGA) contains by far the largest patient cohorts. It is worth noting that given the high dimensions of the whole slide images (see L185), even a single cancer cohort (e.g., TCGA-BLCA) contains over 500GB of data. This is also the case for the benchmarked previously published studies.
>
> **Confidence Intervals:**  We reported standard deviation instead of confidence intervals for two reasons. First, the target variable does not follow a t-distribution ([see this plot](https://filetransfer.io/data-package/djTYYGjo#link)), which is a crucial assumption for the valid interpretation of confidence intervals. As such, reporting standard deviations across the repeated random sub-sampling folds is a more valid estimate of the model’s confidence. Second, using standard deviations was in line with the benchmarked methods, which improves the comparability when directly contrasting the approaches.
>
> **Metrics beyond c-Index:** We agree that Kaplan-Meier plots provide an in-depth representation of the survival probabilities on the test set, but the actual visual differences are often subtle and barely detectible by a naked eye. Therefore, the c-Index coupled with the standard deviation across folds provides a better way of comparing results, as it is also commonly found in the published baselines.
>
> ## **Responses to questions**
>
> **Higher performance reported in MOTCat [1] paper:**
>
> To make the results across experimental setups comparable, we re-ran all experiments under the same experimental conditions and the tunable sets of hyperparameters from the original paper, which we show in Table~1. The different results can be explained by the fact that this paper’s experimental setup is quite different from the one reported in [1]. While [1] predicts disease-specific survival (DSS), HEALNet is trained to predict general survival (GS). General survival does not filter for case deaths unrelated to the disease and is therefore a noisier prediction task. This filtering is evident by the smaller sample sizes on the same cancer cohorts in [1]. Getting higher c-Index results is a general trend we observed in studies predicting DSS instead of GS[2].
>
> HEALNet follows the same experimental setup, evaluation, and cohort selection as in several multimodal benchmarks (Porpoise, MCAT).
>
> **Rationale for weight sharing**:
>
> We would would like to clarify two points: First, the weights are only shared between the sets of modality-specific weights across the fusion layer. For example, the cross-attention update for modality 1 and 2 would be shared across all the fusion layers, but no weights are shared between the modality 1 and 2 within or across fusion layers. Second, the weight sharing helps to 1) improve the parameter efficiency of the model and prevent exploding parameter size with a high number of layers and modalities and 2) we found it to prevent overfitting.
>
> [1] Xu, Y., Chen, H., 2023. Multimodal Optimal Transport-based Co-Attention Transformer with Global Structure Consistency for Survival Prediction, in: 2023 IEEE/CVF International Conference on Computer Vision (ICCV). https://doi.org/10.1109/ICCV51070.2023.01942
>
> [2] Jaume, G., et al., 2023. Modelling dense multimodal interactions between biological pathways and histology for survival prediction. *arXiv preprint arXiv:2304.06819*.

---

> ### Author Response · Authors · 2024-08-13
> **A gentle reminder**
>
> Dear reviewer `a9ao`,
>
> Thank you again for your thoughtful review and positive attitude toward our work! We have addressed all of your questions and concerns. As the discussion period is drawing to a close, we are keen on getting your feedback.
>
> We are looking forward to your response. Thank you for your time.
>
> Kind regards,
> The Authors

---

> ### Author Response · Authors · 2024-08-14
>
> Dear AC,
>
> We really appreciate your time and effort in managing our submission.
>
> Even though we addressed all of reviewer `a9ao` questions and concerns, we've noted that they haven't  responded nor acknowledged our rebuttal thus far, even though they showed positive outlook on our work. We believe further interaction can be beneficial, and we are keen to discuss any remaining clarifications in the final hours of the discussion period.
>
> We would appreciate your assistance in reminding the reviewer to respond/acknowledge our rebuttal.
>
> Kind regards, Authors

---

### Author Rebuttal · Authors · 2024-08-06

### **General comment**

We would like to thank all reviewers for their time and insightful comments. We are encouraged that you acknowledge HEALNet as having an excellent structure and logical flow, sound experiments, and an interesting methodological contribution, as well as being clear, thorough, and unique. We also thank the reviewers for picking up on minor comments and typos which we will incorporate in the final manuscript. In the reviewer comments, we provide detailed answers to your questions with additional explanations of the raised issues.

### **Additional experiments**

We would like to particularly highlight additional experiments that we ran based on the comments around scaling the method to larger datasets (i.e., larger sample size). To address this, we ran HEALNet some relevant baselines on a disease classification (ICD9) and patient mortality (MORT) prediction tasks of MIMIC-III (n=32,616). The table below shows the AUC and Macro AUC for the respective tasks. The modalities in MIMIC are tabular electronic health record features and time series on vital signs and test results.

| Model/Dataset | ICD9 (n=32,616) [AUC] | MORT (n=32,616)  [MacroAUC] |
| --- | --- | --- |
| Uni-modal (EHR) | 0.731 ± 0.023 | 0.658 ± 0.001 |
| Uni-modal (Time Series) | 0.700 ± 0.013 | 0.715 ± 0.016 |
| Perceiver (Early) | 0.733 ± 0.028 | 0.723 ± 0.015 |
| Porpoise (Late) | 0.628 ± 0.020 | 0.617 ± 0.015 |
| **HEALNet** | **0.767 ± 0.022** | **0.748 ± 0.009** |

We did not have time to run all benchmarks given the time and resource constraints during the rebuttal, but we plan to add the full experimental extension to the final manuscript.

---

### Author Response · Authors · 2024-08-12

Dear Reviewers,

Thank you very much for your time and effort in reviewing our paper and responding to our rebuttal.

We would like to encourage those reviewers who are yet to respond to the rebuttal to engage in discussions for the final stretch of the discussion period.

Kind regards,

Authors

---

### Decision · Program_Chairs · 2024-09-25

**Decision:**

Accept (poster)

**Comment:**

The paper presents a method for fusing multi-modal health data for survival prediction in cancer using an iterative cross-attention structure on raw data modalities.
The method can deal with missing modalities. The reviewers point out that this is important in the real-world health data context, where missing variables is a major problem. The method is evaluated on four data sets. The paper has a deep discussion of related work was considered as very detailed.

Similarity to other approaches in the literature and the limitation to survival analysis are mentioned as the main weaknesses.

Yet, after deep discussion with the authors who provided additional results, three out of four reviewers voted towards acceptance of the paper. The one reviewer voting for a weak reject was also the only one to not respond to the rebuttal discussions on the paper.